# Blood Metabolites Associate with Prognosis in Endometrial Cancer

**DOI:** 10.3390/metabo9120302

**Published:** 2019-12-14

**Authors:** Elin Strand, Ingvild L. Tangen, Kristine E. Fasmer, Havjin Jacob, Mari K. Halle, Erling A. Hoivik, Bert Delvoux, Jone Trovik, Ingfrid S. Haldorsen, Andrea Romano, Camilla Krakstad

**Affiliations:** 1Centre for Cancer Biomarkers CCBIO, Department of Clinical Science, University of Bergen, N-5021 Bergen, Norway; Elin.Strand@helse-bergen.no (E.S.); ingvilt@hotmail.com (I.L.T.); Mari.Halle@uib.no (M.K.H.); Erling.Hoivik@uib.no (E.A.H.); Jone.Trovik@uib.no (J.T.); 2Department of Gynaecology and Obstetrics, Haukeland University Hospital, N-5021 Bergen, Norway; 3Mohn Medical Imaging and Visualization Centre, Department of Radiology, Haukeland University Hospital, N-5021 Bergen, Norway; kristine.eldevik.fasmer@helse-bergen.no (K.E.F.); Havjin.Jacob@uib.no (H.J.); ingfrid.helene.salvesen.haldorsen@helse-bergen.no (I.S.H.); 4Section for Radiology, Department of Clinical Medicine, University of Bergen, N-5021 Bergen, Norway; 5Department of Obstetrics and Gynaecology, Maastricht University, 6211 LK Maastricht, The Netherlands; b.delvoux@og.unimaas.nl (B.D.); a.romano@og.unimaas.nl (A.R.); 6ENITEC: European Network for Individualised Treatment of Endometrial Cancer (within the European Society of Gynaecological Oncology), This study is an ENITEC collaboration, https://www.esgo.org

**Keywords:** metabolomics, biomarker, endometrial cancer

## Abstract

Endometrial cancer has a high prevalence among post-menopausal women in developed countries. We aimed to explore whether certain metabolic patterns could be related to the characteristics of aggressive disease and poorer survival among endometrial cancer patients in Western Norway. Patients with endometrial cancer with short survival (*n* = 20) were matched according to FIGO (International Federation of Gynecology and Obstetrics, 2009 criteria) stage, histology, and grade, with patients with long survival (*n* = 20). Plasma metabolites were measured on a multiplex system including 183 metabolites, which were subsequently determined using liquid chromatography-mass spectrometry. Partial least square discriminant analysis, together with hierarchical clustering, was used to identify patterns which distinguished short from long survival. A proposed signature of metabolites related to survival was suggested, and a multivariate receiver operating characteristic (ROC) analysis yielded an area under the curve (AUC) of 0.820–0.965 (*p* ≤ 0.001). Methionine sulfoxide seems to be particularly strongly associated with poor survival rates in these patients. In a subgroup with preoperative contrast-enhanced computed tomography data, selected metabolites correlated with the estimated abdominal fat distribution parameters. Metabolic signatures may predict prognosis and be promising supplements when evaluating phenotypes and exploring metabolic pathways related to the progression of endometrial cancer. In the future, this may serve as a useful tool in cancer management.

## 1. Introduction

Endometrial cancer (EC) is one of the most prevalent cancers among women in Western countries, typically occurring after menopause, and having an increased incidence with advancing age, overweight and obesity [1,2]. This type of gynecological malignancy most frequently presents as endometrioid carcinoma, which usually has a good prognosis when diagnosed at an early stage. Non-endometrioid EC includes high-grade variants, like serous and clear-cell carcinomas, and often has a poorer prognosis.

In general, cancer cells have impaired energy metabolism, and metabolic intermediates typically accumulate in tumors [3]. A deregulated uptake of glucose and amino acids, as well as a utilization of intermediates from glycolysis and the tricarboxylic acid (TCA) cycle, have been considered as major contributors to tumorigenesis [4]. Whereas an altered glucose metabolism is a well-recognized feature of cancer cells, shifts involving fatty acid (FA) metabolism have more recently come into focus and, more specifically, as opposed to in healthy cells, de novo FA synthesis frequently occurs in cancer cells [3].

For patients with EC, there is an urgent need for biomarkers that can help predict the prognosis and, hence, the necessary surgical procedure and personalize the post-surgical care. Currently, only classic histologic classification is used for this purpose, with the aid of some molecular markers, which are, however, used with little consensus among professionals [2]. Due to the association of EC with the metabolic imbalances induced by, e.g., diabetes and obesity, it is likely to assume that metabolic features may be associated with patient prognosis. We have previously demonstrated an association between a high visceral fat percentage and the reduced survival of patients with EC [5]. In addition, as we have shown already for blood steroids [6], plasma metabolites may also be associated with survival and could serve as biomarkers for better prognostication. Metabolomics, the study of metabolites in body fluids and tissues, is a promising tool which has now become more readily available, and has the advantage that it only requires fluids like blood and urine which are easily and non- or minimally-invasively collectable [7]. The aim of the current study was to identify metabolic patterns associated with survival in a selected cohort of women diagnosed with EC. In addition, we aimed to explore whether specific metabolic patterns were associated with the distribution of abdominal fat.

## 2. Results

### 2.1. Cohort Characteristics

The current study included 40 patients with EC, with a median (25th, 75th percentile) age of 72.0 (61.0, 78.5) years and a BMI of 24.0 (23.0, 27.0) kg/m^2^ (Table 1). Overall, patients were followed for median 36.0 (16.5, 66.0) months. Patients with short survival developed recurrent EC during median (25th, 75th percentile) 9.5 (3.50, 14.5) months and died after 16.5 (8.5, 27.0) months, whereas those with long survival were alive and well while being followed for 66.0 (30.0, 70.5) months. There was a significantly higher proportion with myometrial infiltration (>50%) among those with short survival, compared with those having long survival (*p* = 0.03). A total of 15 patients had an endometrioid type of EC, whereas 25 had non-endometrioid type.

### 2.2. Metabolites Associate with Survival

To explore whether blood metabolites or metabolite signatures are associated with patient survival, we performed metabolomics analyses in blood samples collected from a cohort of 40 patients with EC, 20 of whom had long survival, whereas the other 20 had short survival.

The levels of over 180 different molecules including amino acids, biogenic amines, acylcarnitines, hexoses, glycerophospholipids, and sphingolipids were quantified.

Partial least squares discriminant analysis (PLS-DA) was used to detect metabolites which best distinguished cases in relation to survival, and EC patients with short survival could be well distinguished from those with long survival (Figure 1a). The nine most important metabolites which discriminated patients according to survival and having variable importance in projection (VIP) scores >2.0 for component 1 are shown in the VIP heat map (Figure 1b). Results from the cross-validation and permutation are shown in Appendix A.

Next, a hierarchical clustering using the Euclidean distance measure and the Ward algorithm was performed, and 50 features were selected for the sake of visualization (Figure 2). The clustering separated the cohort into two main clusters, with 15 and 25 cases, respectively, but no obvious pattern was demonstrated in relation to survival. Furthermore, there was no significant difference in age or histological type between these clusters, but the fraction of patients with histologic grade 1 was higher in the largest cluster, compared to the smallest (*p* = 0.04).

### 2.3. Metabolite Signature Modeling

With the aim to propose a signature of metabolites that may identify patients with poor prognoses, the following three selection criteria were applied for the inclusion of metabolites based on differences between short and long survival: (1) metabolites with a statistically significant *p* value (<0.05) according to the Mann–Whitney U test; (2) Metabolites with a fold change >2; and/or (3) Metabolites with a VIP score >2.0 based on the PLS-DA.

For further analyses, metabolites with a VIP score >2.0 for components 1 throughout 5 were included in Model 1; metabolites with a VIP score >2.0 for component 1 (but not throughout the other components 2–5) were included in Model 2, together with metabolites from Model 1; and metabolites with a VIP score >2.0 in one or more of components 2–5 (but <2.0 for component 1) were included in Model 3, together with metabolites from model 1 and 2.

Levels of methionine sulfoxide (Met SO, *p* = 0.01) and hydroxypropionylcarnitine (C3-OH, *p* = 0.04) significantly differed according to survival as assessed by the Mann–Whitney U test. None of the metabolites had a fold change >2 and, in addition, C3-OH was slightly under the detection limit (LOD, defined as the value below which metabolite quantification bears some level of uncertainty). Met SO was classified according to the VIP score, and was the only metabolite which fulfilled more than one criterium for the signature. Altogether, the selection criteria resulted in six metabolites in Model 1, ten metabolites in Model 2, and a total of 14 metabolites in Model 3 (listed in Table 2), which we propose as signatures for survival lengths among patients in the current study.

### 2.4. Receiver Operating Characteristics Analyses

Area under the curves (AUCs) were calculated based on receiver operating characteristics (ROC) curves as a measure of model discrimination. When studying the separate metabolites of the proposed metabolic signature according to survival, the AUC was 0.729 (95% CIs 0.564–0.894) for Met SO (*p* = 0.01) and 0.692 (95% CIs 0.525–0.860) for C3-OH (*p* = 0.04). Multivariate ROC analyses containing metabolites of the proposed signature (Models 1–3) yielded an AUC between 0.820 and 0.965 (*p* ≤ 0.001) (Figure 3a–c).

### 2.5. Pathways Involved

To identify pathways which may be differently affected in patients with short, compared to long survival, we performed a joint pathway analysis including transcriptomics data using MetaboAnalyst 4.0 [8]. A significance analysis of microarrays (SAM) was performed on tumor RNA across survival groups in the current patient cohort. All genes which had a fold change >2.0 (*n* = 96), as well as all 14 metabolites which were identified as potentially associated with survival with their corresponding fold changes were included in the analysis. The most affected statistically significant associated pathways were “Mucin type O-Glycan biosynthesis” (*p* = 0.008) and “Retinol metabolism” (*p* = 0.02). These two pathways were also among those showing the strongest pathway topology, in addition to “Linoleic acid metabolism”, “Taurine and hypotaurine metabolism”, and “Cyanoamino metabolism”.

### 2.6. Metabolites Associated with Abdominal Fat Distribution

While body mass index (BMI; kg/m^2^) is a frequently used measure to assess body fat, it is clear that the exact distribution of fat may also affect metabolic pathways in the body [9]. Fat is mainly stored either subcutaneously or viscerally, and can be estimated from computed tomography (CT) images. Previous work by our group has demonstrated the association between a high visceral abdominal fat volume (VAV) percentage and adverse outcomes in patients with EC [5]. The distribution of abdominal fat was measured by CT in a subgroup (*n* = 22) based on the current cohort, and estimates of long and short survival are shown in Table 3. None of the measurements for total abdominal fat volume (TAV, cm^3^), VAV (cm^3^), subcutaneous abdominal fat volume (SAV, cm^3^), VAV percentage, or waist circumference significantly differed between groups with short vs. long survival.

In a PLS-DA, differences according to estimated VAV (cm^3^) and SAV (cm^3^) below and above the median were visualized (Appendix A). VIP scores in Appendix A pinpoint the foremost metabolites which discriminate patients according to values of VAV (cm^3^) and SAV (cm^3^) below or above the median (VIP scores >2.0). Heatmaps of the 50 top-ranked metabolites in patients with (a) VAV (cm^3^) and (b) SAV (cm^3^) values above or below median are shown in Figure 4. These heatmaps indicated that the subjects were clustered into separate groups according to high and low SAV, compared to VAV.

Next, all metabolites which were associated with survival (Table 1) and additional metabolites associated with abdominal fat distribution (Appendix A) were included in Spearman’s ranked correlation plots, illustrating associations between the selected metabolites and the abdominal fat parameters in the subset of patients being examined with CT (Appendix A). Among the strongest correlations are the positive associations between creatinine and all CT-parameters in the long survival group (rho = 0.648–0.830, *p* ≤ 0.04), as opposed to no significant associations in the short survival group. Moreover, there is a strong correlation between (iso)valerylcarnitine with TAV and VAV (rho = 0.830–0.842, *p* ≤ 0.003) in the long survival group, but not in the short survival group. Among patients with short survival, there are significant negative associations for glutamic acid with TAV, VAV, and SAV (rho = −0.580–−0.629, *p* < 0.05), and also strong negative associations for lysoPC-a-C17:0 with TAV, SAV, and waist circumference (rho = −0.804–−0.741, *p* ≤ 0.006).

## 3. Discussion

In the current study, metabolites and metabolic patterns are proposed to be linked with survival in EC, of which Met SO is standing out as one metabolite of particular interest. Unlike previous metabolomics studies related to this type of cancer, we aimed to compare groups of patients with short vs. long survival, despite apparently being matched and similar in other clinical and histological properties. Moreover, associations between metabolites and abdominal fat distribution were explored. A strong association of SAV with specific metabolite profiles was observed, whereas no clear associations could be seen for VAV.

As the most common female cancer, breast cancer has been extensively studied [10,11,12,13,14,15,16,17], also providing untapped opportunities in metabolomics studies in association with gynecological cancers. Moreover, biomarkers have been proposed for application in cervical and ovarian cancers based on metabolomics [18,19,20,21,22,23,24,25].

EC has been less studied in relation to metabolic patterns compared to other gynecological cancers, but recent studies report interesting associations. Several pathways have been suggested to be dysregulated in EC, including the lipid metabolism, by comparing cancer tissue samples with tissue from healthy controls [26]. Moreover, a distinct metabolomic signature of EC has been suggested by an Italian pilot study using different models [27]. A population-based case-control study also identified potential biomarkers in relation to EC which were independent of risk factors like obesity and previously known blood samples, and the authors suggest that metabolic profiling may be important in its early detection and risk assessment [28]. Moreover, metabolomics have demonstrated certain metabolites to be associated with histological subtypes [29,30], with recurrence following surgery [29], and with tumor blood flow in EC patients [31]. A recent study has also evaluated plasma sphingomyelins and phosphatidylcholines as biomarkers in EC [32], proposing ratios between specific metabolites as diagnostic and prognostic models. A diagnostic biomarker signature has also been proposed based on urine metabolomics profiling in EC [33]. A study among subjects with and without EC showed that the metabolic signature differed from both healthy controls, as well as from benign endometrial disease and other gynecological cancers [27].

Altogether, there is an increasing interest in metabolomics related to EC, and for the current study, metabolites involved in amino acid and FA metabolism were mainly investigated. The analyzed metabolites were part of a high throughput kit assay suitable for the analysis of a large number of samples and with a superior reproducibility [34]. Moreover, it requires only a small volume of material for blood analysis. Levels of some metabolites were under the detection limit, which may cause a bias (indicated in Table 2). Analyses were not adjusted for multiple comparisons, which may be a limitation. However, we believe that in the current exploratory study, there may be a higher risk in hiding actual associations due to such adjustments, compared to the risk of revealing false positive results. It is challenging to use PLS-DA in small datasets, as this method is susceptible to increasing the chance of creating an overfitted model. Since the current study does not rely exclusively upon the results from this approach, we believe that PLS-DA is still a useful method for exploring metabolites which seem to determine cohort differences. Moreover, we have based most signature metabolites on a VIP score >2, which is conservative compared to most other studies. ROC curves were made based on several metabolites, and such analysis will provide limited validity in this small study cohort. The findings of the current study should be verified in a larger patient cohort. In contrast to most studies where EC cases have been compared to control, we investigated two matched groups of patients with EC, but with different survival. We did not adjust for potential confounders, which is a limitation. However, the study design in which patients with long survival were matched with those having short survival according to several factors including BMI, was arranged in a way which resulted in adjustment being considered unnecessary. The current findings will thus not necessarily be transferable to a typical population-based cohort of patients with EC. Interestingly, a recent publication based on the same cohort revealed steroid profile as a potential predictor for outcome [6].

Studying properties or activities at the level of genes and proteins has been the most common way of establishing proposed mechanisms in association with health-related research. However, it is challenging to adjust for external factors like nutrition, physical activity, or medications when using this approach. The study of metabolites may be the most promising technique which takes these covariates into account and presents them as a “summary” of all external inputs [35]. Thus, metabolomics can be described as the molecular phenotype or biological end points, which culminates in the response creating the current physiological state, reflecting the other omics, including genomics, transcriptomics, and proteomics [36]. By using metabolomics, one can investigate a large collection of metabolites, which contain several promising biomarkers for risk prediction, diagnosis, and even treatment effects, and end up with certain defined markers related to the outcome of interest, making it very promising for clinical application [37].

Even though most EC cases are successfully treated by initial surgery, 10–15% of the patients experience recurrence within five years with an overall poor outcome [38]. Metabolomics as a prognostic tool may help in identifying patients with an increased risk of recurrence and poor survival, and this will give an advantage, which would enable health services to follow-up specific patient groups.

The metabolome contains a wide range of small endogenous molecules including, but not limited to, amino acids, amines, sugars, lipids, nucleic acids, and FAs, as well as methyl-transfer molecules, each having their own chemical characteristics [39]. Altogether, energy metabolism is affected by several factors, and an impaired metabolism may lead to disease development or affect its progression, which may be reflected through several possible metabolic pathways and components.

As mentioned, Met SO did stand out as one important metabolite, elevated in patients with short survival. Methionine is an essential amino acid and a precursor for succinyl-CoA, homocysteine, cysteine, creatine, and carnitine [40]. Consequently, this amino acid affects the lipid metabolic pathway [41], and changes in methionine cycle may be related to adverse health effects. Met SO is the oxidized form of methionine, and an increased level of this form in the tissues is—under normal circumstances—related with biological aging [42]. It has been demonstrated that the oxidation of methionine impairs the reverse transport of cholesterol through apolipoprotein A-I [43]. Furthermore, a dysfunction in Met SO reductase has been linked with increased cell proliferation, degradation of extracellular matrix, interfering with key signaling processes, and cancer severity [44].

In proliferating cells, nutrients are transformed into intermediates and then used in the biosynthesis of FAs, cholesterol, sugar derivatives like hexose, glycerol, nucleotides, and non-essential amino acids [45]. Important intermediates include one-carbon metabolites, S-adenosylmethionine (SAM), and several metabolites which are involved in glycolysis and the TCA cycle. SAM is a methyldonor involved in epigenetic changes in gene expression, which is generated from the transmethylation pathway from the metabolism of methionine [46]. Moreover, SAM is involved in the synthesis of phosphatidylcholine. The observed higher circulating levels of Met SO in patients with short—compared with long—survival, may reflect a decrease in metabolism through the transmethylation cycle in such patients, and may thus be a potential underlying mechanism causing an impaired cellular function. Interestingly, retinol metabolism was identified as being significantly associated with survival based on the joint pathway analysis. DNA methylation of genes in relation to retinoid signaling is linked to carcinogenesis in several cancers, including EC [47]. Moreover, retinoic acid activates the peroxisome proliferator activated receptor delta, which is central to FA metabolism [48]. Such activation induces oxidative metabolism so that glycolysis is reduced [49].

Hypoxic conditions in tumors lead to a deficiency in the de novo production of unsaturated FAs due to the impaired stearoyl-CoA desaturase 1 induction of double bonds. Instead, unsaturated FAs formed as lysophospholipids are recruited from the surroundings [50]. It has also been demonstrated that metastatic ovarian cancer cells may import FAs from the adipocytes through the upregulation of FABP4 [51]. Overall, tumorigenesis promotes de novo lipid production which leads to changes in membrane composition towards oxidized saturated FAs—an adaptation to oxidative stress [52]. Furthermore, the upregulation of major enzymes involved in biosynthesis of fatty acyl chains is associated with transformed cells [4]. Reactive oxygen species (ROS) may be excessively generated in proliferating cells due to accumulated electron transport flux, and can activate transcription factors like HIF1α and NRF2 making further contributions to tumorigenesis [4].

Previous work by our group has demonstrated a high VAV percentage to be associated with adverse outcomes in patients with EC [5]. Based on these findings we performed a subgroup analysis including on those in the current study who also had CT data available. Interestingly, a defined pattern appears to be present when stratifying the cohort into extents of VAV and SAV, respectively, with a particularly well-defined pattern for SAV. Notably, abdominal subcutaneous adipose tissue has been demonstrated to increase in women up till 60–70 years, and is overall more insensitive to weight reduction compared to visceral fat [53]. Several metabolites correlate with SAV, which seems to predict prognosis to a lesser extent than VAV, being compliant with the weak associations between metabolites and survival. The data on abdominal fat distribution, as measured by CT, adds interesting perspectives to the current study, although the number of included patients is low.

Overall, being a small study cohort, the findings would not necessarily be reflected in a similar larger cohort or at the population level. However, we believe that the current study illustrates that metabolomics is a promising future approach when exploring potential biomarkers for clinical use.

## 4. Materials and Methods

### 4.1. Study Population

All samples were retrieved from The Bergen Gynecologic cancer biobank (REK 2014/1907). The biobank includes samples prospectively collected from patients diagnosed with gynecological cancer in Western Norway between 2001 and 2015 and is a population-based patient series. Included patients gave written informed consent. The biobank has been thoroughly described previously [54]. A subgroup of 20 patients with short survival time and having either FIGO (International Federation of Gynecology and Obstetrics 2009 criteria) stage 1 and grade 1–3, or having stage 2 and grade 2 diseases were selected from this cohort. From the same cohort, 20 patients with long survival were matched with those having short survival for FIGO stage, histology, grade, age, body mass index (BMI), and parity, and were selected for the analyses.

The study was performed in accordance with the principles of the Declaration of Helsinki and was approved by the Regional Committee for Medical Research Ethics (REK 2009/2315).

### 4.2. Metabolomic Profiling

EDTA blood was obtained before primary surgery. The samples were centrifuged at 1600× *g* for 15 min and the plasma was stored at −80 °C until analysis was performed. Non-fasting plasma samples were analyzed on a multiplex platform which included 183 metabolites (Biocrates: AbsoluteIDQ^®^ p180 kit assay, Innsbruck, Austria). The complete metabolomics data were produced at the contract research department of Biocrates (Innsbruck, Austria) according to the company protocol [55]. In short, the samples were cleared by centrifugation and then analyzed. The automated assay was based on PITC (phenylisothiocyanate)-derivatization in the presence of internal standards. For the quantification of acylcarnitines, lipids, and hexose, flow-injection-analysis (FIA)-MS/MS was used, where 20 μL of sample was directly injected into the mass spectrometer via electrospray ionization using an AB SCIEX 4000 OTrap^®^ mass spectrometer (AB SCIEX, Darmstadt, Germany). Amino acids and biogenic amines were analyzed by LC/MS. The samples (10 μL) were injected on a reverse-phase column and the column eluent was introduced directly into the mass spectrometer by electrospray ionization and using an AB SCIEX 4000 OTrap^®^ mass spectrometer (AB SCIEX, Darmstadt, Germany). For each metabolite, the limit of detection (LOD) was calculated as three-times the median of the concentrations in the zero sample for the metabolite under consideration.

### 4.3. Transcriptomics

RNA was extracted from primary tumor tissue using the RNeasy Mini Kit (Qiagen, Germantown, MD, USA). Microarrays [5,56] and RNA-sequencing were performed as previously described [6].

### 4.4. Image Analysis on CT Scans

Diagnostic abdominal preoperative CT scans were performed as routine diagnostic procedures and were available in 22 patients. An assessment of abdominal fat was performed by a semi-automated method for volumetric quantification (iNtuition, TeraRecon Inc., San Mateo, CA, USA) [57], from which the visceral (VAV) and subcutaneous (SAV) abdominal fat volumes were estimated (cm^3^). The total abdominal fat volume (TAV) was then considered as the sum of VAV and SAV. Percentage of visceral abdominal fat volume (VAV%) was then calculated ([VAV/TAV] × 100; VAV%). Waist circumference was measured in an axial image at the L3/L4 level (iNtuition, TeraRecon Inc., San Mateo, CA, USA).

### 4.5. Statistical Analyses

Medians (25th, 75th) and counts (%) were reported for the continuous and categorical variables, and differences between groups were calculated using the Mann–Whitney U test for continuous variables and Chi-square for independence on binary variables.

Log-transformed data were imported into the MetaboAnalyst 4.0 software (Xia Lab, McGill University), in which *t*-tests were performed for metabolite levels between specified groups and fold change was detected. PLS-DA was used to detect variable importance in projection (VIP) scores. Metabolites which significantly differed as calculated by the Mann–Whitney U test, which had a fold change threshold of 2, and/or a VIP value >2.0, were selected for the proposed metabolic signature according to survival. Moreover, heatmaps were generated by hierarchical clustering using the Euclidean distance measure and the Ward algorithm, based on the top 50 metabolites which differed most according to the group in question.

We explored model discrimination by calculating areas under receiver operator characteristics curves (ROC–AUC) on both separate components, as well as on metabolic signatures. For subgroup analysis, Spearman-ranked correlation was used to illustrate the association between selected metabolites and abdominal fat distribution parameters. Furthermore, the 22 patients with available preoperative CT data were stratified according to high vs. low VAV and SAV, as defined by each of their median [5].

Statistical analyses were performed using IBM SPSS Statistics for Windows, version 24 (SPSS, Armonk, NY, USA), R version 3.2.5 (R Development Core Team, Vienna, Austria), and MetaboAnalyst 4.0 [58]. Probability values were two-sided and considered statistically significant when *p* < 0.05.

## 5. Conclusions

This study revealed that metabolic signatures are associated with survival and abdominal fat distribution in EC. Thus, metabolomic profiling stands out as a robust approach in prognostic assessment. In the future, metabolomics profiling may serve as a useful tool for identifying targets in treatment as well as patient groups in need of close follow-up.

## Figures and Tables

**Figure 1 metabolites-09-00302-f001:**
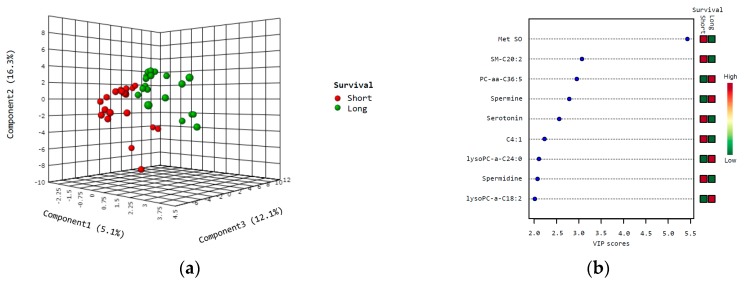
PLS-DA and variable importance in projection (VIP) scores in survival analysis. Partial least square discriminant analysis (PLS-DA) was used to identify metabolites which could separate cases according to short (red) vs. long (green) survival (**a**); VIP heat map, showing metabolites with VIP scores > 2.0 (based on component 1) (**b**). Abbreviations: C4:1, butenylcarnitine; Met SO, methionine sulfoxide.

**Figure 2 metabolites-09-00302-f002:**
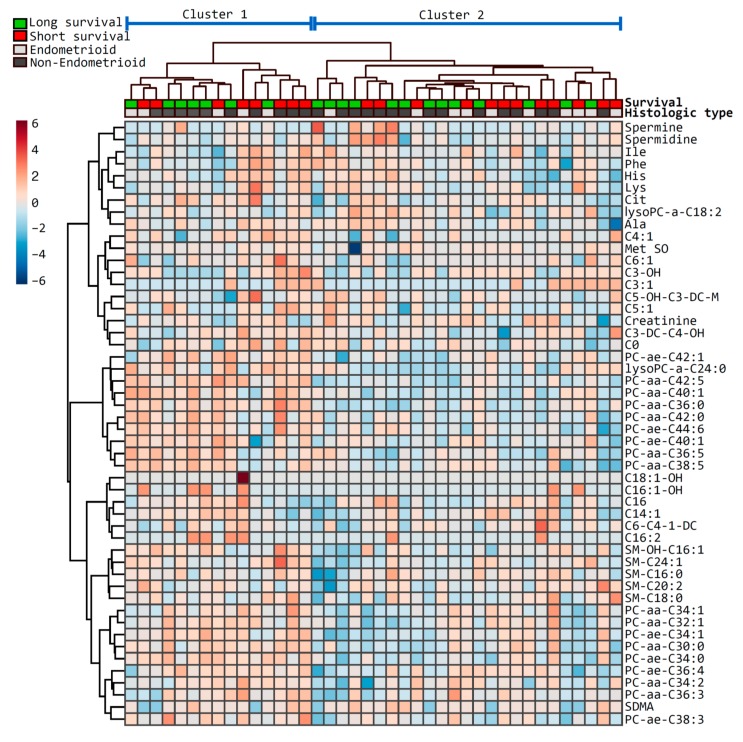
Heatmap of the top metabolites differentially presented in patients with short and long survival. The heatmap illustrates the most relevant metabolites ranked by *t*-test according to short (red) and long (green) survival as defined on top. Histologic type is indicated below the survival (endometrioid, light grey; non-endometrioid, dark grey). Metabolites are clustered along the vertical axis, whereas subjects are clustered along the horizontal axis (main clusters are indicated on top).

**Figure 3 metabolites-09-00302-f003:**
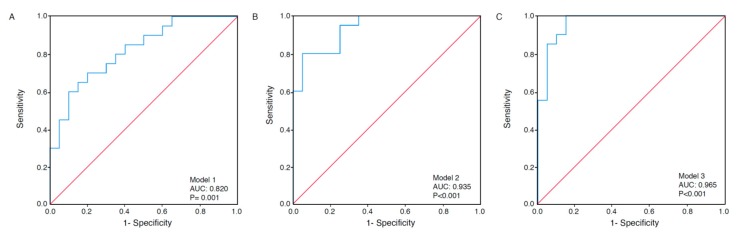
Receiver operating characteristics (ROC) curves under three modeling methods. Multivariate ROC analyses based on the metabolites in Models 1–3, according to survival. Model 1 is based on the metabolites methionine sulfoxide (Met SO), serotonin, spermine, hydroxypropionylcarnitine (C3-OH), PC-aa-C36:5, and SM-C20:2 (AUC 0.820, 95% CIs 0.692–0.948) (**A**). Model 2 is based on all metabolites in Model 1, as well as spermidine, butenylcarnitine (C4:1), lysoPC-a-C18:2, and lysoPC-a-C24:0 (AUC 0.935, 95% CIs 0.865–1.000) (**B**). Model 3 is based on all metabolites in Models 1 and 2, as well as aspartic acid (Asp), asymmetric dimethylarginine (ADMA), hexose H1, and PC-ae-C30:1 (AUC 0.965, 95% CIs 0.913–1.000) (**C**). *p* values are based on the asymptotic 2-tail significance.

**Figure 4 metabolites-09-00302-f004:**
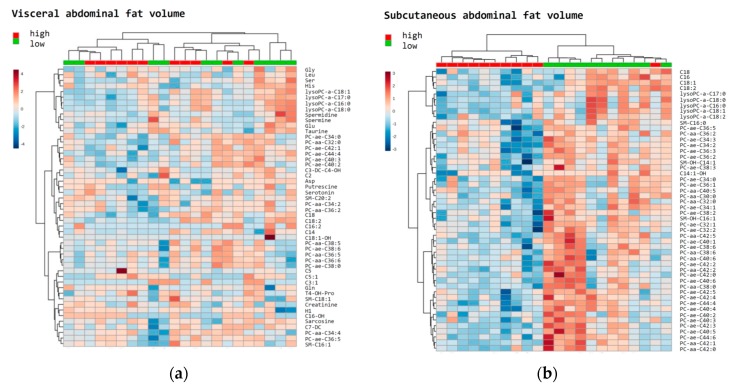
Heatmaps of the top metabolites in relation to fat distribution. The most relevant metabolites ranked by *t*-test were used to discriminate a total of 22 patients according to high (red: above median) vs. low (green: below median) visceral abdominal fat volume (cm^3^) (**a**); and subcutaneous abdominal fat volume (cm^3^) (**b**).

**Table 1 metabolites-09-00302-t001:** Clinico-pathological characteristics of included patients.

Characteristics	Total Cohort (*n* = 40)	Long Survival (*n* = 20)	Short Survival (*n* = 20)	*p* ^†^
Age (years)	72.0 (61.0, 78.5)	67.0 (56.0, 77.0)	75.0 (63.5, 81.5)	0.10
Body Mass Index (BMI, kg/m^2^)	24.0 (23.0, 27.0)	24.0 (22.0, 26.5)	26.0 (23.0, 27.0)	0.20
Recurrence Free Survival (months)	28.5 (9.5, 66.0)	66.0 (60.0, 70.5)	9.5 (3.50, 14.5)	<0.001
Recurrence, *n* (%)	21 (52.5)	1 (5.0)	20 (100)	<0.001
Follow-up Time (months)	36.0 (16.5, 66.0)	66.0 (60.0, 70.5)	16.5 (8.5, 27.0)	<0.001
Myometrial Infiltration ≥50%, *n* (%)	19 (47.5)	6 (30.0)	13 (65.0)	0.03
Histologic Type, *n* (%)				
Endometrioid Type	15 (37.5)	7 (35.0)	8 (40.0)	0.74
Non-endometrioid Type				
Clear Cell	3 (7.5)	3 (15.0)	0 (0)	0.07
Serous Papillary	8 (20.0)	3 (15.0)	5 (25.0)	0.43
Carcinosarcoma	11 (27.5)	6 (30.0)	5 (25.0)	0.72
Other non-endometrioid	3 (7.5)	1 (5.0)	2 (10.0)	0.55
Histologic Grade, *n* (%) ^#^				
Grade 1	6 (15.0)	3 (15.0)	3 (15.0)	1.00
Grade 2	4 (10.0)	2 (10.0)	2 (10.0)	1.00
Grade 3	5 (12.5)	2 (10.0)	3 (15.0)	1.00
FIGO Stage, *n* (%)				
Stage I	36 (90.0)	18 (90.0)	18 (90.0)	1.00
Stage II	4 (10.0)	2 (10.0)	2 (10.0)	1.00

Values are presented as medians (25th, 75th percentiles) or counts (percentages), ^†^ Calculated by using the Mann–Whitney U test for continuous variables and Chi-square for independence on binary variables, ^#^ Shown for endometrioid cases only.

**Table 2 metabolites-09-00302-t002:** Plasma metabolites associated with survival among 40 patients with endometrial cancer.

Metabolite	Total Cohort (*n* = 40)	Long Survival (*n* = 20)	Short Survival (*n* = 20)	*p* ^†^	VIP Score ^#^
***Amino Acids and Biogenic Amines (µM)***					
Asp ^3^	7.1 (5.9, 8.5)	6.8 (6.1, 7.9)	7.6 (5.8, 9.1)	0.76	5.32
ADMA ^3^	0.55 (0.30, 0.70)	0.50 (0.30, 0.80)	0.60 (0.30, 0.65)	0.90	2.93
Met SO ^1^	1.20 (1.00, 1.50)	1.05 (0.90, 1.25)	1.40 (1.10, 1.55)	0.01	5.43
Serotonin ^1^	0.75 (0.45, 1.25)	0.65 (0.45, 1.35)	0.90 (0.45, 1.25)	0.78	2.67
Spermidine ^2^	0.30 (0.30, 0.40)	0.30 (0.20, 0.40)	0.40 (0.30, 0.40)	0.17	2.08
Spermine ^1^**	0.20 (0.20, 0.30)	0.20 (0.20, 0.30)	0.20 (0.20, 0.30)	0.62	2.79
***Acylcarnitines (µM)***					
C3-OH ^1^**	0.027 (0.023, 0.030)	0.026 (0.023, 0.028)	0.028 (0.025, 0.034)	0.04	1.63
C4:1 ^2^**	0.023 (0.018, 0.026)	0.022 (0.017, 0.025)	0.023 (0.020, 0.027)	0.22	2.23
***Sugar (µM)***					
Hexose H1 ^3^	4051 (2915, 4868)	3776 (2915, 4714)	4098 (3014, 5385)	0.75	2.62
***Glycerophospholipids and Sphingolipids (µM)***					
lysoPC-a-C18:2 ^2^	27.0 (21.5, 35.6)	32.1 (22.3, 36.0)	25.3 (18.7, 35.3)	0.29	2.01
lysoPC-a-C24:0 ^2^	0.38 (0.20, 0.54)	0.42 (0.20, 0.59)	0.36 (0.20, 0.52)	0.33	2.11
PC-aa-C36:5 ^1^	39.4 (27.7, 56.9)	42.7 (38.2, 57.0)	30.2 (24.4, 55.0)	0.07	2.95
PC-ae-C30:1 ^3^	0.026 (0.00, 0.12)	0.025 (0.00, 0.098)	0.026 (0.00, 0.15)	0.69	2.16
SM-C20:2 ^1^	0.59 (0.43, 0.69)	0.57 (0.33, 0.65)	0.60 (0.47, 0.77)	0.16	3.07

Values are presented as medians (25th, 75th percentiles), ^†^ Calculated by using the Mann–Whitney U test; ^#^ The highest VIP score based on components 1–5; ^1^ Model 1: methionine sulfoxide (Met SO, HMDB02005), serotonin (HMDB00259), spermine (HMDB01256), hydroxypropionylcarnitine (C3-OH, HMDB13125), PC-aa-C36:5 (HMDB07890), and SM-C20:2 (HMDB13465); ^2^ Model 2: all metabolites in Model 1, spermidine (HMDB01257), butenylcarnitine (C4:1, HMDB13126), lysoPC-a-C18:2 (HMDB10386), and lysoPC-a-C24:0 (HMDB10405); ^3^ Model 3: all metabolites in Model 1 and 2, aspartic acid (Asp, HMDB00191), asymmetric dimethylarginine (ADMA, HMDB01539), hexose H1 (HMDB00143), and PC-ae-C30:1 (HMDB13402); HMDB names are given according to the The Human Metabolome Database: http://www.hmdb.ca/metabolites. ** Those metabolites marked with double asterisks had values that fell under the detection limits, therefore, though they could be measured, the prediction may not be accurate.

**Table 3 metabolites-09-00302-t003:** Abdominal fat estimates from pre-operative computed tomography according to survival among 22 patients with endometrial cancer.

Abdominal Fat Estimates	Total Cohort (*n* = 22)	Long Survival (*n* = 10)	Short Survival (*n* = 12)	*p* ^†^
TAV (cm^3^)	6933 (5654, 8746)	7131 (6534, 8746)	6758 (5373, 8683)	0.64
VAV (cm^3^)	2388 (1920, 3916)	2666 (2086, 3461)	2219 (1905, 3919)	0.74
SAV (cm^3^)	4151 (3329, 5389)	4437 (3916, 5923)	3980 (3263, 5252)	0.55
VAV%	37.4 (33.4, 43.5)	37.1 (31.3, 40.3)	38.0 (35.0, 45.7)	0.45
Waist Circumference (cm)	93.2 (86.2, 99.1)	91.6 (86.0, 95.9)	97.3 (87.0, 99.2)	0.34

Values are presented as medians (25th, 75th percentiles). ^†^ Calculated by using the Mann–Whitney U test. Abbreviations: SAV: subcutaneous abdominal fat volume; TAV: total abdominal fat volume; VAV: visceral abdominal fat volume; VAV%: visceral fat percentage.

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
