# Peer review of "Blood Metabolites Associate with Prognosis in Endometrial Cancer"

_metabolites, 2019, doi:10.3390/metabo9120302_

Round 1

Reviewer 1 Report

The authors provide a well written and presented manuscript. The overall impact of the work is moderate, as the numbers used are small and the authors have not really presented any viable leads for robust biomarkers which could be used for predictive modelling of cancer prognosis. 

There a few general comments that need addressing before this manuscript can be considered for publication.

The authors consistently refer to the "top 50 ranked metabolites". This seems are extremely arbitrary selection. The authors should use valid and robust selection criteria with clear thresholds of feature importance. This arbitrary selection of variables would indicate a lack of overall explanatory features within the dataset. Why has correction for multiple testing not been used (ie, bonferroni, FDR).This is essential for determining the validity and robustness of potential biomarkers. Line 125-126; The authors state that C2-OH was slightly under the detection limit yet still quantifiable. Limit of Detection it typically lower than te Limit of Quantification. It seems strange that a compound is < LoD but > LoQ. C3-OH should be marked as < LoQ and not used in analysis. In Table 1 of the 14 biomarkers presented; only 3 a are statistically significant. Why such a focus on negative results ? Figure 4; The x axis label is missing Table 2; none of this data is significant. The authors should consider it is actually relevant to the main manuscript text. Materials & Methods; 4.3 and 4.2: There is a severe lack of detail here. The authors must vastly improve these sections and provide detailed methodology which would allow their findings to be properly reproduced.

The overall findings of the manuscript are relatively weak, and the results presented do not strongly support their aim of developing biomarkers for prognosis of endometrial cancer. 

Author Response

Reviewer 1:

The authors provide a well written and presented manuscript. The overall impact of the work is moderate, as the numbers used are small and the authors have not really presented any viable leads for robust biomarkers which could be used for predictive modelling of cancer prognosis. 

There a few general comments that need addressing before this manuscript can be considered for publication.

The authors consistently refer to the "top 50 ranked metabolites". This seems are extremely arbitrary selection. The authors should use valid and robust selection criteria with clear thresholds of feature importance. This arbitrary selection of variables would indicate a lack of overall explanatory features within the dataset.

Response: The “top 50 ranked metabolites” were not meant to be a main focus of this manuscript. These were merely used to gain a proper number of metabolites included when illustrating that patients may be clustered according to certain patterns of metabolites. We do not believe that the reference to such “top 50 ranked metabolites” interferes with the overall findings of this study as these are only used to visually illustrate possible associations between clusters and outcome in Figures 2 and 5 (currently Figures 2 and 4). We have now modified the original sentence on page 4, lines 107-108, into a more straightforward explanatory text when presenting the first of these two figures illustrating the mentioned data: “Next, a hierarchical clustering using Euclidean distance measure and the Ward algorithm was performed, and 50 features were selected for the sake of visualization (Figure 2).”

Why has correction for multiple testing not been used (ie, bonferroni, FDR).This is essential for determining the validity and robustness of potential biomarkers.

Response: As this is an exploratory approach aiming to reveal specific metabolic patterns, and not really biomarkers per se, correction for multiple testing was not considered essential for the current study. A review by KJ Rothman argues the common use of adjusting for multiple comparisons [1]. Adjusting for multiple comparisons reduces the chance of type I errors. However, this is at the expense of increasing the chance of gaining type II errors. We have now included a few sentences in the manuscript addressing the lack of adjustment for multiple comparisons in the current study (page 9-10, lines 239-242): “Analyses were not adjusted for multiple comparisons, which may be claimed as a limitation. However, we believe that in the setting of an exploratory approach as in the current study, there may be a higher risk in hiding actual associations due to such adjustment, compared to the risk of revealing false positive results.”

However, your comment may imply that the title of this manuscript is misleading, and we have therefore chosen to remove the word biomarker from the title, leaving the revised title as follows: “Blood metabolites associate with prognosis in endometrial cancer”

Line 125-126; The authors state that C2-OH was slightly under the detection limit yet still quantifiable. Limit of Detection it typically lower than te Limit of Quantification. It seems strange that a compound is < LoD but > LoQ. C3-OH should be marked as < LoQ and not used in analysis.

Response: In our analyses, we set only the LOD for each metabolite, and we defined such LOD as the value below which the metabolite quantification is less trustable than in case values are above the LOD. This means that in our definition, LOD is based on the linearity of the standard, and are not values that corresponded to the absence of a peak (LOQ), rather to peaks whose integration was not robust. We did not really define a LOQ. In case of C3-OH, the LOD is 0.05, but peaks could be still be integrated and measured (real measurements vary between 0.0166 and 0.0466, mean 0.027). Technically, LOD is calculated as three-times the median of the concentrations in the zero sample for each metabolite in the standard curve. We have amended the text to clarify this issue (page 5, line 135) and added in the methods the description of the LOD (page 12, lines 341-343).

In Table 1 of the 14 biomarkers presented; only 3 a are statistically significant. Why such a focus on negative results ?

Response: As explained in section 2.3, metabolites which were selected to be included in the metabolic signature were based on 1) having a P-value <0.05 according to the Mann-Whitney U test, 2) having a fold change >2, or 3) having a VIP score >2.0 (based on PLS-DA). Based on these criteria, metabolites were further classified into 3 models, an approach which is common within epidemiology to assure robust methodology. We appreciate this important comment by the reviewer, and would like to point out that the selection of metabolites does not exclusively depend on a statistical significant P-value, since in our opinion, such approach would underestimate the importance of several metabolites.

Figure 4; The x axis label is missing

Response: We agree that Figure 4 does not clearly illustrate the findings. We have therefore decided to remove the figure and have only included the main findings from the joint pathway analysis in the text on page 7, lines 164-168.

Table 2; none of this data is significant. The authors should consider it is actually relevant to the main manuscript text. 

Response: Despite that there are no P-values <0.05 in Table 2 (now Table 3), we believe it is relevant to show these results. Even though there are no statistical significant differences in survival across the various adipose tissue parameters as measured by CT, metabolite patterns seem to be associated with adipose tissue composition, particularly with SAV.

Materials & Methods; 4.3 and 4.2: There is a severe lack of detail here. The authors must vastly improve these sections and provide detailed methodology which would allow their findings to be properly reproduced.

Response: We thank the reviewer for the comment and are happy to provide more information if needed.

The overall findings of the manuscript are relatively weak, and the results presented do not strongly support their aim of developing biomarkers for prognosis of endometrial cancer. 

Response: The aim of the current study was to identify metabolic patterns and not necessarily biomarkers per se, as previously discussed. In our opinion, the aim has been properly addressed by the findings in this study, based on the proposed metabolic signature. As previously noted, we have modified the title to better reflect the aim and findings of the study.

Reviewer 2 Report

In this paper the authors describe a study in which a metabolite signature (blood plasma) for Endometrial Cancer (EC) is sought.

The study consists of performing metabolmics analysis on 40 samples consisting of 20 with good and 20 bad prognosis.

Finding a signature that correlated with prognosis is clearly a useful goal.

I have the following concerns with the paper:

There seems to be little statistical consideration taken in the analysis. 183 metabolites are measured, and some seem to correlate with prognosis. PLS can find dimensions that split the classes. Neither is necessarily a surprise, and so some statistical analysis (with correction for multiple testing) is required to determine if what is being seen is significant. This issue goes through the paper. The signature is chosen based upon various measures of difference between the two groups. It is therefore unsurprising that the AUCs are quite high. Although the only solution to this problem is generating more samples (which is presumably tricky), I think the authors should make the limitation of this evaluation much clearer. Figure 4: the caption is not detailed enough. What do "enrichment" and "homology" mean in this context. Fig 5: this shows the most relevant metabolites ranked by t-test. Again, not clear what has been done to account for multiple testing, and also whether or not these are significant, or just the lowest p-values? Why are both Mann-Whitney and t-test used? 

Author Response

Reviewer 2:

In this paper the authors describe a study in which a metabolite signature (blood plasma) for Endometrial Cancer (EC) is sought.

The study consists of performing metabolmics analysis on 40 samples consisting of 20 with good and 20 bad prognosis.

Finding a signature that correlated with prognosis is clearly a useful goal.

I have the following concerns with the paper:

There seems to be little statistical consideration taken in the analysis. 183 metabolites are measured, and some seem to correlate with prognosis. PLS can find dimensions that split the classes. Neither is necessarily a surprise, and so some statistical analysis (with correction for multiple testing) is required to determine if what is being seen is significant. This issue goes through the paper.

Response: Being an exploratory study, we have clearly stated the chosen approaches made during this study. Please see the response to Reviewer 1 regarding correction for multiple testing.

The signature is chosen based upon various measures of difference between the two groups. It is therefore unsurprising that the AUCs are quite high. Although the only solution to this problem is generating more samples (which is presumably tricky), I think the authors should make the limitation of this evaluation much clearer.

Response: We agree and thank the reviewer for pointing this out. We have added a sentence regarding this issue in the discussion on Page 9-10 (lines 242-243): “ROC curves were made based on several metabolites, and such analysis will provide limited validity in this small study cohort.”

Figure 4: the caption is not detailed enough. What do "enrichment" and "homology" mean in this context.

Response: Please see response to reviewer 1, addressing the same issue.

Fig 5: this shows the most relevant metabolites ranked by t-test. Again, not clear what has been done to account for multiple testing, and also whether or not these are significant, or just the lowest p-values? Why are both Mann-Whitney and t-test used? 

Response: We believe that in this case, a non-parametric test like the Mann-Whitney U test provides a robust test, and the metabolites included in the signature is thus based on this test. The t-test is the method lying behind the heatmap which was merely used to illustrate the visual patterns like in Figure 5 (now changed into Figure 4). Please also see the responses to Reviewer 1, who addresses similar issues.

Reviewer 3 Report

This is an interesting manuscript addressing an extremely important issue, the identification of biomarkers for endometrial cancer. I have a number of comments and suggestions for the authors.

One of the biggest limitations of this study is the small sample size; although this is not unexpected given the prevalence of the cancer and the availability of suitable blood samples, this should be addressed.

The main analytical method also has limitations that must be considered by the authors, notably the issue of overfitting. Furthermore, the fact that this does not take any kind of confounding into account, such as BMI, which the authors themselves note could be a confounder, again the issue of confounding should be addressed.

 The authors method of generating ROC curves in the same population based on VIP score, cannot be considered as validation of their score of evidence of its robustness, as such the authors should be cautious in their conclusions.

It seems little troubling that in the three models, all of which are based on the same data and all of which are trying to do the same thing, are comprised of almost entirely different metabolites (as I read it only one is common to all three) how do the authors explain this? Furthermore, Section 2.2 seems to describe 6 different models, but uses the terms model 1,2 and 3, twice. Please clarify how metabolites were chosen for each model, for example is model 2 based on fold change, or on component 1, or on both?

The authors should provide more detail on what the aim of section 2.5 was, and more interpretation of the results

Minor comments

In the abstract, FIGO should be defined from the outset

In the first paragraph of the results please define ‘good prognosis’ and ‘short survival’ and ‘long survival’.

Table 3 showing patient characteristics, should be Table 1 in the results section, and details on matching should be provided here to help contextualize the results

In section 2.2 the authors should clarify that the p-values they report for Met SO and C3-OH are based on the Mann-Whitney U test, this is not clear from the text

All AUCs should be presented with 95% confidence intervals.

In Figure 3, the p-values shown for each graph should be explained.

In the pathway analysis section the authors should state how many genes were included. Are these genetic results published elsewhere? Further information on this would be helpful.

Section 2.55, define VAV; TAV; SAV and CT

Author Response

Reviewer 3:

This is an interesting manuscript addressing an extremely important issue, the identification of biomarkers for endometrial cancer. I have a number of comments and suggestions for the authors.

One of the biggest limitations of this study is the small sample size; although this is not unexpected given the prevalence of the cancer and the availability of suitable blood samples, this should be addressed.

Response: We thank you for this comment. These patients were selected based on a relative difference in survival during three years. We agree that the patient cohort should be larger. However, it was not possible to include more patients locally without compromising the inclusion criteria. We strongly emphasize in our discussion that our results need validation in a larger cohort (page 10, lines 243-244).

The main analytical method also has limitations that must be considered by the authors, notably the issue of overfitting. Furthermore, the fact that this does not take any kind of confounding into account, such as BMI, which the authors themselves note could be a confounder, again the issue of confounding should be addressed.

Response: Confounding is always important when evaluating epidemiological data, and we agree that this issue should be addressed. We have now included a short discussion regarding confounding factors, including BMI (page 10, lines 246-249): “We did not adjust for potential confounders, which is a limitation. However, the study design in which patients with long survival were matched with those having short survival according to several factors including BMI, was arranged in a way which resulted in adjustment being considered unnecessary.”

The authors method of generating ROC curves in the same population based on VIP score, cannot be considered as validation of their score of evidence of its robustness, as such the authors should be cautious in their conclusions.

Response: The same issue was addressed by Reviewer 2, in which we agree. We have added a sentence regarding this issue in the discussion on Page 9-10 (lines 242-243), as replied to reviewer 2.

It seems little troubling that in the three models, all of which are based on the same data and all of which are trying to do the same thing, are comprised of almost entirely different metabolites (as I read it only one is common to all three) how do the authors explain this? Furthermore, Section 2.2 seems to describe 6 different models, but uses the terms model 1,2 and 3, twice. Please clarify how metabolites were chosen for each model, for example is model 2 based on fold change, or on component 1, or on both?

Response: We agree that this should have been described more clearly. There are not 6 models, but there are 3 selection criteria and 3 models. Model 1 is the “basic” model, Model 2 includes all metabolites from Model 1, but 4 additional, and Model 3 includes all metabolites in Models 1 and 2, but 4 additional. We have now rephrased the text to clarify how metabolites were selected for each model. Please see page 4, lines 114-123.

The authors should provide more detail on what the aim of section 2.5 was, and more interpretation of the results

Response: We agree with this comment and have added an introduction to Section 2.5 (now renumbered to 2.6) regarding these analyses (page 7, lines 171-175).

Minor comments

In the abstract, FIGO should be defined from the outset

Response: This has now been done.

In the first paragraph of the results please define ‘good prognosis’ and ‘short survival’ and ‘long survival’.

Response: This paragraph has now been rephrased to provide explainable definitions.

Table 3 showing patient characteristics, should be Table 1 in the results section, and details on matching should be provided here to help contextualize the results

Response: We have now moved the entire section regarding Cohort characteristics from the Methods to the Results section of the manuscript, including Table 3, which has now been named Table 1. Table 1 and 2 have been named Table 2 and 3, respectively.

In section 2.2 the authors should clarify that the p-values they report for Met SO and C3-OH are based on the Mann-Whitney U test, this is not clear from the text

Response: We have already stated in the text the following (page 5, lines 132-133): “Levels of methionine sulfoxide (Met SO, P=0.01) and hydroxypropionylcarnitine (C3-OH, P=0.04) significantly differed according to survival as assessed by the Mann-Whitney U test.”

All AUCs should be presented with 95% confidence intervals.

Response: All 95% confidence intervals have now been presented in Section 2.4.

In Figure 3, the p-values shown for each graph should be explained.

Response: A sentence has been added to the figure legend (page 7, line 157): “P values are based on the asymptotic 2-tail significance.”

In the pathway analysis section the authors should state how many genes were included. Are these genetic results published elsewhere? Further information on this would be helpful.

Response: We have now included the number of genes which were used in the pathway analyses (n=96) (page 7, line 162). These genetic data was also included in [2].

Section 2.55, define VAV; TAV; SAV and CT  

Response: These abbreviations have now been appropriately defined.

Round 2

Reviewer 1 Report

The authors still haven’t provided sufficient detail in the Material & Methods. In particular, within 4.2 Metabolomic Profiling there should be enough information here so that another researcher could reproduce their analysis. The authors have not provided even basic details on LC run conditions (ie, mobile phase, gradient, temperature) and likewise for MS operating conditions. The authors should review recent submissions in MDPI metabolites; such as https://doi.org/10.3390/metabo9110271 to see the level of detail that is expected.

The authors have provided some extra detail and clarity in the manuscript which improves the understanding of some key elements. The change in title also now better reflects the content of the manuscript.

My main concern with the manuscript is still that the foundation of the results, are extremely weak. Inclusion of confidence intervals for AUC values (Figure 3) are very welcome, and highlight where there are both strengths and weaknesses across their different models.

Overall the authors have provided a substantially improved manuscript. The authors still need to improve the level of detail in the Materials and Methods (as mentioned above), and they should emphasise in the discussion the preliminary/pilot nature of their results. The authors should make the reader more aware of the consequence of the low power combined with limited statistical significance, within the discussion.

Author Response

Reviewer 1:

The authors still haven’t provided sufficient detail in the Material & Methods. In particular, within 4.2 Metabolomic Profiling there should be enough information here so that another researcher could reproduce their analysis. The authors have not provided even basic details on LC run conditions (ie, mobile phase, gradient, temperature) and likewise for MS operating conditions. The authors should review recent submissions in MDPI metabolites; such as https://doi.org/10.3390/metabo9110271 to see the level of detail that is expected.

Response: The complete metabolomic data was produced at the contract research department of Biocrates according to the company’s protocol. It is policy of the company not to disclose the full details of the protocol, and previous studies published using Biocrates service for metabolomics indeed report basic info on the protocol and refer to the patent Ramsay and co-workers - US 2007/0004044 (ref 56 of our study). See as examples on the implementation of this policy (Lehmann, Friedrich et al. 2015, Rotroff, Corum et al. 2016, His, Viallon et al. 2019).

We contacted Biocrates directly who strongly recommended to follow this same procedure for our study as well. 

We have amended the M&M to include in any case more info on the methods, but this does not include gradients and phases used for elution. If the Reviewer and the Editor believe that it is fundamental to include this info in the methods, we will approach again Biocrates.

The following information has now been added to Section 4.2 of the manuscript (page 12, lines 338-347): “The complete metabolomic data was produced at the contract research department of Biocrates according to the company protocol. In short, samples were cleared by centrifugation and then analyzed. The automated assay was based on PITC (phenylisothiocyanate)-derivatization in the presence of internal standards. For the quantification of acylcarnitines, lipids, and hexose, flow-injection-analysis (FIA)-MS/MS was used, where 20μL of sample were directly injected into the mass spectrometer using via  electrospray ionization using AB SCIEX 4000 OTrap® mass spectrometer (AB SCIEX, Darmstadt, Germany). Amino acids and biogenic amines were analyzed by LC/MS. Samples (10 μL) were injected on a reverse-phase column and the column eluent was introduced directly into the mass spectrometer by electrospray ionization and using an AB SCIEX 4000 OTrap® mass spectrometer (AB SCIEX, Darmstadt, Germany).“

The authors have provided some extra detail and clarity in the manuscript which improves the understanding of some key elements. The change in title also now better reflects the content of the manuscript.

My main concern with the manuscript is still that the foundation of the results, are extremely weak. Inclusion of confidence intervals for AUC values (Figure 3) are very welcome, and highlight where there are both strengths and weaknesses across their different models.

Overall the authors have provided a substantially improved manuscript. The authors still need to improve the level of detail in the Materials and Methods (as mentioned above), and they should emphasise in the discussion the preliminary/pilot nature of their results. The authors should make the reader more aware of the consequence of the low power combined with limited statistical significance, within the discussion.

Response: We thank the reviewer for pinpointing these issues and as we realize that even though findings will not appear strong in such a small cohort, they illustrate that metabolomics is a promising approach when exploring potential biomarkers for clinical use. We have now addressed this more clearly in the discussion on page 11, lines 318-320: “Overall, being a small study cohort, the findings can not necessarily be reflected in a similar larger cohort or at the population level. However, we believe that the current study illustrates that metabolomics is a promising future approach when exploring potential biomarkers for clinical use.”

References:

His, M., V. Viallon, L. Dossus, A. Gicquiau, D. Achaintre, A. Scalbert, P. Ferrari, I. Romieu, N. C. Onland-Moret, E. Weiderpass, C. C. Dahm, K. Overvad, A. Olsen, A. Tjonneland, A. Fournier, J. A. Rothwell, G. Severi, T. Kuhn, R. T. Fortner, H. Boeing, A. Trichopoulou, A. Karakatsani, G. Martimianaki, G. Masala, S. Sieri, R. Tumino, P. Vineis, S. Panico, C. H. van Gils, T. H. Nost, T. M. Sandanger, G. Skeie, J. R. Quiros, A. Agudo, M. J. Sanchez, P. Amiano, J. M. Huerta, E. Ardanaz, J. A. Schmidt, R. C. Travis, E. Riboli, K. K. Tsilidis, S. Christakoudi, M. J. Gunter and S. Rinaldi (2019). "Prospective analysis of circulating metabolites and breast cancer in EPIC." BMC Med 17(1): 178.

Lehmann, R., T. Friedrich, G. Krebiehl, D. Sonntag, H. U. Haring, A. Fritsche and A. M. Hennige (2015). "Metabolic profiles during an oral glucose tolerance test in pregnant women with and without gestational diabetes." Exp Clin Endocrinol Diabetes 123(7): 483-438.

Rotroff, D. M., D. G. Corum, A. Motsinger-Reif, O. Fiehn, N. Bottrel, W. C. Drevets, J. Singh, G. Salvadore and R. Kaddurah-Daouk (2016). "Metabolomic signatures of drug response phenotypes for ketamine and esketamine in subjects with refractory major depressive disorder: new mechanistic insights for rapid acting antidepressants." Transl Psychiatry 6(9): e894.

Reviewer 2 Report

The authors have gone some way to addressing my comments, but I still have reservations.

I don't agree with the comment that multiple testing correction is not appropriate in an exploratory study. The goal of an exploratory study must be to see if there is a relationship between any metabolites and the conditions of interest. The way in which they are doing this is showing that there are metabolites with p<0.05. However, there will always be such metabolites by chance, and therefore multiple testing is vital! If nothing is significant after multiple testing then the result of the exploratory analysis is that there is no evidence of a relationship! I feel that these results must be included.

Author Response

Reviewer 2:

The authors have gone some way to addressing my comments, but I still have reservations.

I don't agree with the comment that multiple testing correction is not appropriate in an exploratory study. The goal of an exploratory study must be to see if there is a relationship between any metabolites and the conditions of interest. The way in which they are doing this is showing that there are metabolites with p<0.05. However, there will always be such metabolites by chance, and therefore multiple testing is vital! If nothing is significant after multiple testing then the result of the exploratory analysis is that there is no evidence of a relationship! I feel that these results must be included.

Response: We do not claim that correction for multiple testing is not appropriate in an exploratory study in general, however we did not consider it as essential for the current study. We realize that in any statistical approach, there will be chance findings. Moreover, in the current study the cohort and design was special in many ways, and it is noteworthy that the two groups were matched for several traits. However, the main statistical methodology used in this manuscript has been founded on the packages delivered by Metaboanalyst, and we have based most analyses on these established methods. The VIP scores which define most of the signature metabolites are not based on P values, but scores the contribution of a variable to the model. Whereas the typical “cut-off” for selecting relevant variables is usually set to a VIP score greater than 1, we already decided to set this level to greater than 2 due to the small sample size and low power. Overall, we have chosen statistical methods which, in our opinion, seem to explore the current aim in a way which is appropriate. To underline that we are aware of this we have added some sentences about this in the discussion on page 9-10 (lines 240-244): “Analyses were not adjusted for multiple comparisons, which may be a limitation. However, we believe that in the current exploratory study, there may be a higher risk in hiding actual associations due to such adjustment, compared to the risk of revealing false positive results. Moreover, we have based most signature metabolites on a VIP score >2, which is conservative compared to most other studies.

Reviewer 3 Report

I am satisfied that the authors have addressed all my comments; and made favorable edits to the manuscript elevating its overall significance

Author Response

Reviewer 3:

I am satisfied that the authors have addressed all my comments; and made favorable edits to the manuscript elevating its overall significance

Response: We are grateful to the reviewer for pointing out important issues during the review, providing us a chance to improve the manuscript.

Round 3

Reviewer 2 Report

The authors have added a couple of sentences to their report which are reasonable.

I still disagree re the multiple testing: the claim the paper is making is that there looks to be a signature that distinguishes the two groups. That's the answer to a statistical question and therefore any answer ought to be backed up with statistics.

It is certainly of interest, in the context of that question, to know whether or not there are any significant metabolites after correction. I understand that OPLS-DA is a different approach (one that will also give peaks high coefficients by chance) and a legitimate one, but Mann-Whitney p-values are quoted in the results tables and, as they haven't been corrected, they are misleading.

In their response, the authors claim that the signature is chosen by VIP > 2, although this statement in the methods seems to contradict that suggesting MW p-values are also used? Although perhaps this is a typo, as the sentence doesn't really scan?

Metabolites which significantly differed as calculated by the Mann-Whitney U test, which had a fold change threshold of 2, and/or a VIP value >2.0, were selected for the proposed metabolic signature according to survival.

Finally, no details are provided on the PLS-DA procedure? Does the implementation in Metaboanalyst make use of some kind of cross-validation procedure to optimise the VIP parameters? 

Author Response

As we respect the Reviewer’s opinion regarding the multiple testing, the statistics using Mann Whitney U test is a non-parametric choice, which is frequently used when evaluating such data. However, the proposed signature was evaluated on the composite basis of differences measured with the Mann-Whitney U test, of differences based on a fold-change threshold of 2, and/or a VIP value >2. Overall, as illustrated in Table 2, only one of the metabolites which satisfied the overall criteria had a VIP value <2, and that is the acylcarnitine C3-OH.

We have now added results from the cross validation and permutation test as also commented by the Editor. A comprehensive protocol describing MetaboAnalyst 4.0 has recently been published (Chong, Wishart et al. 2019).

REFERENCES

Chong, J., D. S. Wishart and J. Xia (2019). "Using MetaboAnalyst 4.0 for Comprehensive and Integrative Metabolomics Data Analysis." Curr Protoc Bioinformatics 68(1): e86.
